# Medical Fraud and Abuse Detection System Based on Machine Learning

**DOI:** 10.3390/ijerph17197265

**Published:** 2020-10-05

**Authors:** Conghai Zhang, Xinyao Xiao, Chao Wu

**Affiliations:** 1School of Management, Zhejiang University, Hangzhou 310058, China; 21851193@zju.edu.cn; 2School of Material Science and Engineering, Qingdao University of Science and Technology, Qingdao 266042, China; x0chaoren0s@gmail.com

**Keywords:** healthcare fraud, medical abuse, anomaly detection

## Abstract

It is estimated that approximately 10% of healthcare system expenditures are wasted due to medical fraud and abuse. In the medical area, the combination of thousands of drugs and diseases make the supervision of health care more difficult. To quantify the disease–drug relationship into relationship score and do anomaly detection based on this relationship score and other features, we proposed a neural network with fully connected layers and sparse convolution. We introduced a focal-loss function to adapt to the data imbalance and a relative probability score to measure the model’s performance. As our model performs much better than previous ones, it can well alleviate analysts’ work.

## 1. Introduction

In China, the health care market is worth more than 2 trillion yuan per year. Besides the considerable market, Chinese national health care security administration randomly inspected 197 thousand of medical institutions and found that nearly 1/3 of them had existing health care violations in 2018, China [1]. It is estimated that approximately 10% of healthcare expenditure is wasted due to health care abuse or fraud behavior, which makes it an essential issue for health care systems.

Usually, inappropriate healthcare behavior includes system error, medical abuse, and healthcare fraud [2]. With the development of health care settlement system, the settlement process gets more faultless, but still it cannot prevent intentional deception.

Generally, medical abuse means that healthcare service providers offer unnecessary medical treatments or services to the patient, to get more profit or kickbacks. Healthcare fraud is an intentional deception used, which is intended to obtain unauthorized benefits [2]. Usually, it is implemented by an intentional “patient” or a group of malicious “patient” rather than a medical service provider, which gets more complex to supervise. In recent days, it was reported that a family pretended to be sick for dozens of diseases and asked for hundreds of pills per day, and it turned out that they had frauded the healthcare funds for 400 thousand in a year. [3] Since healthcare fraud is more harmful to health insurance funds, the laws of most countries/regions define it as illegal. Both healthcare fraud and the milder medical abuse damage the health insurance system and finally result in the harm of social welfare.

Medical abuse and healthcare fraud behaviors are slightly different from typical medical behaviors, as a professional data analyst can discriminate between the abnormal suspect and regular records by reviewing multiple dimension records. However, as the participation rate is more than 95% and the daily records go over 100 thousand, facing such a massive number of records, data analysts cannot comprehensively review all of the records [4].

With the help of machine learning, we can train a model to classify the abnormal records by learning samples’ characteristics. In order to find fraudulent behaviors, there are several main difficulties to deal with:(1)There is no exact rule which can clearly distinguish the abnormality of medical insurance transactions. Moreover, the number of abnormal records is tiny compared to the massive number of regular treatment records. For those two reasons above, the relatively small dataset of labeled abnormal records limit the algorithm accuracy.(2)Due to the influence of various concomitant diseases, patient characteristics, doctor preferences, and additional noise factors in medical treatment records, the situation is complicated, making the anomaly challenging to find out [5].(3)Because intentional deception fraudsters often use multiple methods to conceal their fraudulent behaviors behind enormous usual transaction data, traditional means based on rules are challenging to find fraudsters and hard to cover the updated fraud behaviors.(4)The frequent changes upon the medical insurance drugs list or disease relations, call for in time updating the logic of anomaly detection. Additionally, as a result of massive data, it will take much time to perform retraining or redetection to update the anomaly detection system.

For those reasons listed above, the real-world healthcare scenario is so complex that many reasonable behaviors seem abnormal, and hence the abnormal detection system in the healthcare domain is known as hard to develop and apply. In order to get rid of this dilemma, we tried to use a machine learning method (neural network) to detect medical fraud cases. The results proved that our model can indeed significantly alleviate the analysts’ work.

In our work, we used neural networks to understand the combination of disease and prescription, which plays a significant role in medical abuse or fraudulent behavior. After the construction of feature engineering, we applied an outlier detection model to find suspicious anomaly records. In the last place, medical data analysts re-checked those suspicious records and made analysis. The result of the experiment shows that our model can improve the discover rate of abnormal health care behavior.

## 2. Related Work

Traditionally speaking, in the healthcare anomaly detection area, machine learning applications can be divided into supervised learning and unsupervised learning. Typically, supervised learning requires data-label for training, while unsupervised learning does not.

Supervised learning application algorithm applied in anomaly detection includes neural network classification, genetic algorithms, support vector machines, decision trees, KNN, etc. [6]. Supervised learning algorithms generally get better performance in detecting known patterns of fraud and abuse than the unsupervised learning algorithms, which are usually partitioning, agglomerative, or probabilistic, etc. [7].

However, a supervised learning algorithm depends heavily on datasets [8]. As the healthcare scenario is complicated, a dataset is usually not comprehensive, which often causes the result to be seriously over-fitted in real-world scenarios. For example, in the literature [9], through the study of thousands of diabetes data, they used 9 financial features to build a decision tree and obtained 99% abnormal behavior detection rate on the experimental data set. The anomaly detection system implemented in [10] implements the classification based on the local labeled data set which was idealized. Literature [11] uses the Bayesian network in anomaly detection on simulated data, proving the feasibility of the Bayesian network in the field of medical insurance anomaly detection. The real environment is often more complicated than the simulated data or partial data, so those algorithms are hard to be implemented.

The unsupervised method theoretically needs to prove the consistency problem of outliers and anomaly. Literature [12] analyzes the dental outpatient records in Australia. The anomaly of local outliers was studied and found that part of the data anomalies is caused by direct or indirect misuse or fraud behavior, which proves the feasibility of outlier detection in medical insurance abnormal behavior detection.

Therefore, unsupervised learning is currently widely used for abnormal data identification of medical insurance transactions. The main methods of unsupervised learning include clustering algorithms such as K-means and Dbscan, algorithms based on probability density function estimation, and others using the isolation forest approach.

In recent years, people have made a series of new research in this field. R. Ikono et al. (2019) [13] reviewed 88 articles from journal articles, conference minutes, and books based on the research question’s relevance. The results of this review indicate that traditional fraud detection methods were difficult to be implemented in the healthcare system, as new fraud patterns continue to evolve to circumvent fraud detection methods.

Part of the emerging research focuses on constructing a complex framework that can describe the relationship between multiple entities (such as patients, doctors, services, etc.), and hopes to dig out the ins and outs of medical insurance fraud. Ekin, Tahir et al. (2019) [14] proposed a hierarchical model to help the investigators’ group medical procedures and identify the hidden patterns among providers and medical procedures. Irum Matloob et al. (2020) [15] proposed a framework to computes association scores for three entities (patient, doctor, service), and use G-means clustering over the scores to predict whether a case is a fraud or not.

In contrast, our paper proposes a new model to identify outliers from a simple disease–drug relationship, so as to directly find medical insurance fraud in prescription records. The type of data required by our new method is simple, so the amount of data required is relatively less, and the prescription record contains less privacy and is easier to obtain than the record of the relationship between patients, doctors, and services. Moreover, as the information is deeply hidden in dozens of features, which was challenging to use, those algorithms’ effectiveness was poor, and the researches are hard to be implemented to the real scenery. We sacrificed the precision for a recall trade-off to solve these problems. Therefore, our model is more practical.

### 2.1. Entities and Data Claim

As in the digital health care system in Zhejiang Province, China, the database contains encrypted personal information, medical information, payment settlement information, and so on, mainly as Table 1:

We got about 7.37 million encrypted treatment records beginning from 2014 on 300,000 people sampled from Hangzhou, Zhejiang, China.

Treatment records have several characteristics as below:(1)The sample number varies from disease to disease; the number of treatment records for some rare diseases is one hundred thousandth of the number of records for the most common diseases.(2)Diseases might be a combination of primary disease, chronic disease, and concurrent disease, which will change the relationship between the disease and prescription.(3)Both the number of disease categories and the number of drug categories are more than 20 thousand.(4)Due to the differences between different regions and hospitals’ built-in systems, the content’s coding methods or even databases have various structures, causing the record’s format to be inconsistent and invalid.(5)Performance optimization design can result in lots of separated data tables, complex relationship connections, and redundant relationships.

For those reasons listed above and the data unevenness, we limit the problem’s complexity by constraining the categories of disease and drug. However, although we constrained the drug and disease categories into the most common 1000th, there is a severe distribution unevenness in the dataset, as the top 1 disease has nearly 1.2 million treatment records, while the 1000th disease only has only 10 thousand records. Figure 1 shows the top 10 diseases with records count, which indicates that chronic diseases get higher incidence than others.

We used the adjustment factor to tune the down sample and up sample rate. Each disease’s target sample count will be the original count multiplying the factor, which is intended to approach the 1000 diseases’ average records count. For those rare diseases with factor larger than 10, we clipped the factor under 10 to avoid overfit in the model. The top 10, 100, 500, 1000 diseases’ numbers are shown in Figure 2.

### 2.2. Modelling

The features we designed was mainly based on analysts’ experiences, including personal information, history health-care behavior, and so on. Among feature parts we generate the features as Table 2. In order to get better result during outlier detection, we filtered the features into 20 features depending on mutual information base on 1000 manual labeled records.

One of the most important features we needed was the disease–prescription relation score, as the relation was hard to be quantified, we designed a multi-label classification model to quantify the relation between 0–1.

First of all, considering the large number of categories and possibilities by the combination, we finally clipped the disease and drug categories into the most common 1000th, which cover the 71% records. In this way, the rest of 29% records containing uncommon disease/drug cannot be detected, but as the anomaly detection task is low-recall, the sacrifice for better learning of disease–drug combinations can improve the overall recall. After necessary data cleaning for missing and invalid, we got about 5 million available treatment records. During the preprocessing, we transferred those original records into formatted features, encoded those features, applied one-hot code to the disease and prescription, and then overlaid disease’s or drug’s one-hot vectors to present the combination of concurrent diseases or drugs.

We proposed a multi-label relative classification method using a neural network with fully connected layers and sparse convolution for better comprehension. Because the output medicine correlates with each other, rethink [16] architecture was applied before the output layer, which improves the model’s performance. The whole network mainly consists of 7 layers. The first layer fully connects the input into 512 dimensions. The next two layers are convolution layers that encode the input from 512 dimensions into 16 × 16 × 8 and 4 × 4 × 32. The next three layers convert it back into 1000 dimensions before the output rethink rnn was applied.

We use the Formula (1) to calculate the loss of multi-label classification, *y_k_* means the ground truth in feature *k*, and $\sigma \left(l_{k}\right)$ means the sigmoid of output *l_k_*_._ However, for the reason that the distribution of 1000-dimension output vector tends to be zero, we used focal loss for the equalization, which were calculated as Formula (2), in which α is used to adjust the imbalance and γ called the focusing parameter, which is set to 2.0 in our work in order to increase the loss of difficult features:(1)$$\mathrm{Loss}=-\;\overset{m}{\underset{k}{\sum }}\left[y_{k}log\left(\sigma \left(l_{k}\right)\right)+\left(1-y_{k}\right)log\left(1-\sigma \left(l_{k}\right)\right)\right]$$
(2)$$FocalLoss=-\alpha_{t}{\left(1-p_{t}\right)}^{\gamma }\log(p_{t})$$

Because the influence between output labels is relatively small, training was performed as a sigmoid multi-label training. However, the actual score was based on relative probability and then calculated the score as formula below, which is the average probability of ground truth label. In Formula (3), *y_k_* means the predicted label of feature *k* among all feature *m*, and *y_gt_k_* means the ground-truth of each feature.
(3)$$p=\frac{\sum_{k}^{m}y_{k}y_{gt\_k}}{\sum_{k}^{m}y_{k}}$$

We tested this relationship score model using the random split test set. We tested several algorithms, including multilabel decision tree (ML-DT), rank SVM, as showed in Table 3, using the neural network gets lowest one-error, which indicates the comprehensive to this relationship was best. The average probability of the sum of 1000 features of drugs in a single sample in the test set was 0.006. On the contrast, the average probability calculated by the formula above is 0.453, which means the target’s drug got more average likelihood than other drugs (the likelihood is hundreds of times of others). Compared with the average probability of 1000 classes, it indicated that the model had already obtained the disease–drug relationship, and there was no obvious overfitting of positive and negative samples.

Then we used the formula above to obtain the correct prediction score of each record, which stands for the disease–drug relationship score, then combined it with personal information, historical transaction information, and other features to perform outlier detection. The feature generation step can be concluded in Figure 3.

In order to generate as much as features to find out what might be related to the medical abuse or fraudulent behavior, based on the advice of analysts, we manually designed some quantitative characteristics derivative from the original data such as personal payment ratio, medical fund usage limit, personal credit score, and cumulative outpatient visit count.

After generating about 12 other features using feature combination, along with the relationship score we used covariance and mutual information scores to explain the features’ correlation. Then we remove redundant and no-obvious features, and finally got the most relevant features: medical insurance pay rate, patients’ out-of-pocket payment rate, disease–prescription match score, frequency of outpatient visit, hospital location, and patient credit score.

As is shown in Figure 4, we found that in our dataset, the accuracy of clustering (K-means and Dbscan) is much lower than the isolation forest [17] method.

## 3. Results

Because there were no available labeled datasets the same as Zhejiang province’s health care system, we randomly sampled a dataset from the database and then labeled it, which contained 100 known abuse/fraud records and another 900 known regular records.

We tested the result using the traditional rule and sorts, and it only gets 24% among the top 100 records. In contrast, we tested several outlier detection algorithms using the features generated in Figure 3, the result as Table 4.

The traditional methods mainly include percentiles and other statistical factors, then sort the final weight-sum score and get the most abnormal records. The rest of the algorithm was based on the normalized features generated as in Table 2 in the previous section. In order to simulate the anomaly detection in the real-world (which usually checks the specific rate of all samples due to the massive data), we used metrics called Detection rate (DR), which indicates the actual anomaly samples counts in the top 10% of overall samples. K-means and DBscan were based on the most common Euclidean distance and got a 35% and 33% detection rate. We find that the Isolation Forest does the best detection among the top 100 records, which got 47% accuracy in the top 100 scores. The Isolation Forest algorithm intends to build several trees which split the similar samples and gather similar samples. As a result of few available samples, we set the estimators in the forest to 100, and the sample number was 256 on whole features with a random shuffle. We believed that the Isolation Forest was better in searching and dealing with irrelevant attributes, which resulted in the highest score. As Figure 5 shows, the Isolation Forest based on the comprehensive features with disease–drug relation scores gets a higher detection rate than traditional means.

We unsampled the 1000 records with 10× SMOTE on both normal/abnormal records for a better test of our detection model, tested K-means, and Isolation Forest on this 10 k SMOTE dataset. The result was shown in Table 5:

In the SMOTE 10 k dataset, the Isolation Forest using the same parameters (estimators and sample number) gets a lower detection rate compared to the original 1 k dataset. However, the K-means gets higher in contrast, which may indicate that the distribution tended to be a global anomaly.

Among the first 100 records (with highest relative probability scores), our model detected 55 actual abnormal ones, which means that healthcare data analysts only need to check the top 100 records scored by our model and will find the 55-abnormal behavior. By contrast, 550 of that or even more (according to the abnormal rate) would be needed to check without our model. In conclusion, our method can concentrate the abnormal records and help save the manual check effort.

After the above experimental steps on the original 7.37 million anonymized healthcare records, we obtained suspected abnormal records under the abnormal rate of one ten-thousandth. Since the whole records were too numerous to examine, we sampled 500 consecutive samples every 20% of data order by abnormal score then made manual labeling, and we found that the relationship between the abnormality and the actual abnormal ratio is as shown in Table 6, where the number of abnormal behaviors is more noticeable than origin data.

Our anomaly detection system is not only effective but also efficient. The feature extraction module takes 25 min @ 1080ti for training on 7 million treatment records. Retraining is needed only when there are large-scale changes in drug/disease. Additionally, the inference time complexity corresponds to the number of samples linearly, about 1 min @ 5 million data. The outlier detection time complexity is linear also.

## 4. Discussion

During the process of manual labeling, we found that different data analysts have different concepts of abnormal and they are constrained by their domain knowledge of disease or prescription, which will result in the inconsistency of results.

There are several patterns which can be found in the final abnormal records:

1.Drug dosage abuseIn some cases, during the treatment, there exist much more drug dosages than usual, and usually, the drug belongs to a local pharmaceutical factory and gets a high price.

2.Duplicate testA patient takes the generally unnecessary same test multiple times. For example, in one of the abnormal records, a patient was charged 3 times for testing antibody to hepatitis B surface antigen (HBsAb) by a medical service provider in a single treatment.

3.Unrelated drugsMedical provider provides unrelated drugs, which will show low disease/drug relation score in the record. For example, one of the Chinese patent drugs for trauma was applied to a patient with influenza.

4.Unrelated serviceSeveral medical services are frequently happened in health care abuse, like medical massage aimed for pulling muscle but used in cold treatment.

5.Drugs with similar effects abuseSome records appear that excessive drugs and dosages with similar effects were used for a single treatment. Usually, it relates to medical providers’ abuse.

6.Excessive outpatient frequencySome records show that the patient does excessive outpatient behaviors. For example, a malicious user visited different hospitals on average twice a day, and every time he asked the doctor for pseudoephedrine hydrochloride prescriptions.

While reviewing the abnormal data, we found that the records whose prescription includes Chinese traditional patent medicines made by the local pharmaceutical factory are more likely to be abusive. Meanwhile, we noticed that the abnormal rate in pharmacies and private hospitals is higher than that of official public hospitals, which may indicate that those places need to be specially supervised.

In conclusion, those main kinds of abnormal records have a commonality that the summary price of treatment is more than other records, whose average amount was five times more than regular records. The overall average amount for a single treatment is 300 yuan per outpatient record. We found that the most typical of those abnormal behaviors is drug dosage abuse, and usually, the drug dosage abuse behavior’s drug has a high price, which directly related to the profit or kickback.

## 5. Conclusions and Future Work

We provided a model that can detect abnormal records in the healthcare area, using outlier detection and an end-to-end multi-label prediction method for disease–prescription correlation scores. The most significant advantages of this model are that it requires a simple data type, provides admirable practicability, achieves better accuracy and recall compared to the traditional rule-based method, which alleviates data analysts’ work.

When we tried to improve the performance of the model, we restricted the range of data as we limited the number of drugs and diseases to less than 1000. As a side-effect of accuracy improvement, this anomaly detection system can only detect about 71% of records. Due to the unlabeled dataset’s constraints and the massive amount of data, we can only verify the sampled result and check on our 1000 records dataset or do level-sample tests, which are not comprehensive enough to cover every disease and drug.

Meanwhile, the limitation of our paper is that due to privacy and security issues involved, we failed to obtain enough training data thoroughly. The medical history is incomplete, and the dataset contains serious noise even though we do data clean for several times. The results contain misjudgment and can only be an assistant tool for data analysts for efficiency. We need to endure more effort to ask for access to databases stored in government, hospitals, and relative institutions as it is playing a pivotal role in building a perfect model. On the other hand, in order to balance the skewed label classes, we tried the adjustment factor to tune the downsample and upsample rate, which produced an acceptable result. However, there remain more techniques to enhance the performance, such as SMOTE, threshold moving, or transform the supervised classification problem to an unsupervised anomaly detection problem.

## Figures and Tables

**Figure 1 ijerph-17-07265-f001:**
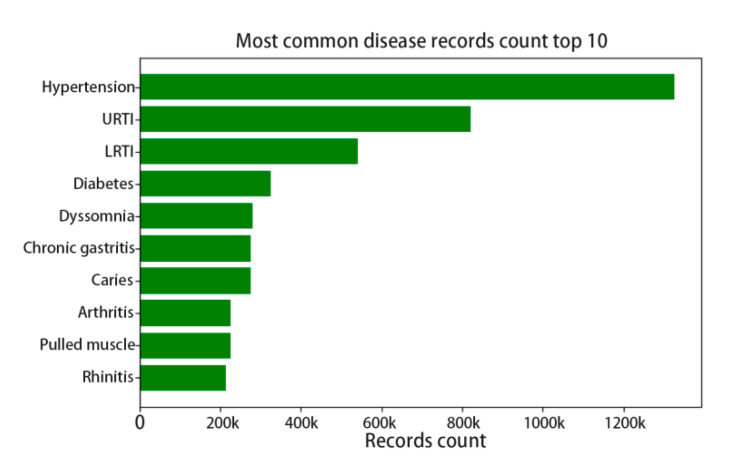
The top 10 disease record count distributions.

**Figure 2 ijerph-17-07265-f002:**
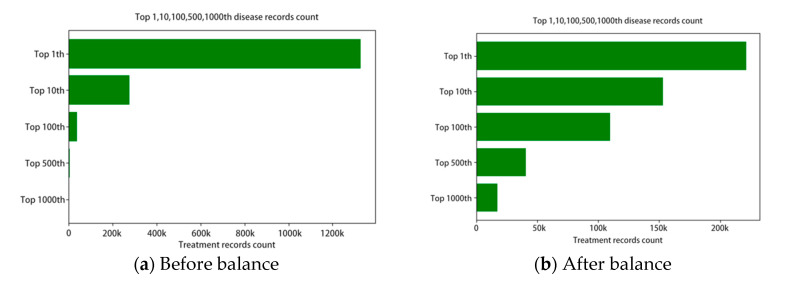
The disease record count distributions.

**Figure 3 ijerph-17-07265-f003:**
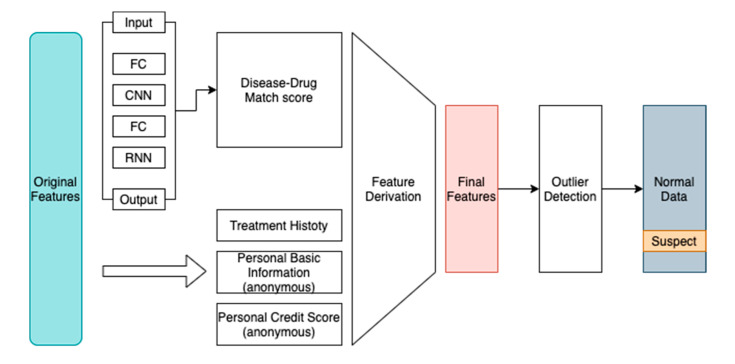
System diagram.

**Figure 4 ijerph-17-07265-f004:**
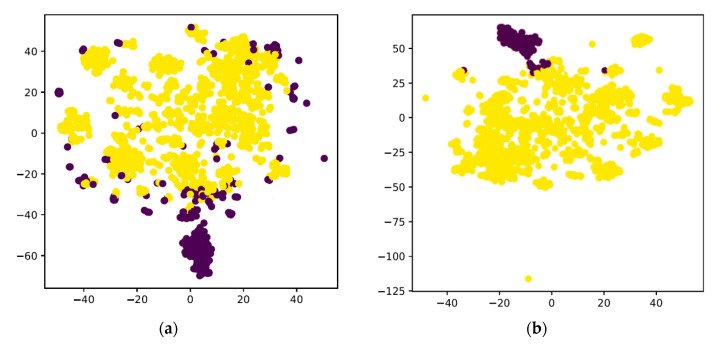
(**a**) the t-SNE result by K-means; (**b**) the result by isolation forest.

**Figure 5 ijerph-17-07265-f005:**
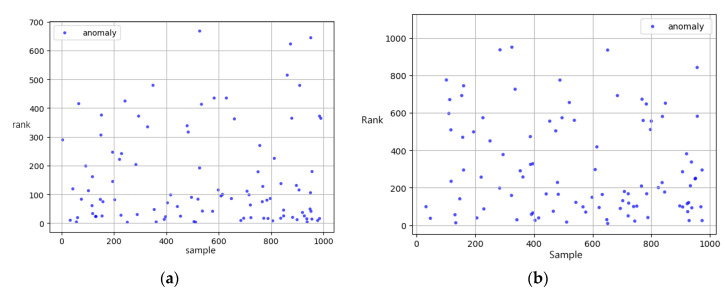
Ranks of known abnormal record based on (**a**) our model (**b**) traditional rule.

**Table 1 ijerph-17-07265-t001:** Main data structure in the digital healthcare system.

Personal Attributes	Payment Detail	Settlement Detail	Hospitalized Detail
Person ID	Detail-ID	Settlement-ID	Hospitalized-ID
Sex	Person ID	Person ID	Person ID
Age	Insurance pay	Total cost	Type
Medical insurance usage	Drug name	Type of insurance	Department code
Outpatient amount	Number of medications	Medical insurance costs	Hospitalized days
Hospitalized amount	…	…	…

**Table 2 ijerph-17-07265-t002:** Main features.

Dimension	Attributes	Description
Fee	Total amount	Total amount during last year
	Total health-care pay	Total health-care fund paid amount during last year
	Total self-pay	Total health-care fund paid amount during last year
	Total amount of medicine	Total cost of medicine during last year
	Average amount	Average amount during last year
	Average self-pay	Average self-pay during last year
	Average medicine fee	Average medicine fee during last year
	Maximum self-pay	Maximum self-pay during last year
	Total amount percent in all patients …	Rank percent sort by total amount in all patients
Frequency/Hospital	Total visit times	Total hospital visits times last year
	Average gap	Average gap between hospital visits
	Hospital count	Total visited hospital count last year
Personal information	Age	
	Health-care type	
	Gender	
	Total balance	
	…	Other description
Treatment detail	Primary disease	
	Secondary disease	
	Prescription	Prescription drugs list
	Maximum amount of single drugs	Maximum amount of single drugs prescription

**Table 3 ijerph-17-07265-t003:** Multilabel algorithms’ result.

Algorithm	One-Error	Coverage
ML-DT	0.670	25.801
Rank SVM	0.733	36.962
NN	0.427	13.431

**Table 4 ijerph-17-07265-t004:** Detection rate of different algorithms.

Algorithm	Detection Rate (DR)
Traditional rule sorts	24.0%
K-means	35.0%
DBScan	33.0%
Isolation Forest	47.0%
LocalOutlierFactor	45.0%

**Table 5 ijerph-17-07265-t005:** Detection rate of different algorithms on the SMOTE 10 k dataset.

Algorithm	Detection Rate (DR) @10%
K-means	38.1%
Isolation Forest	45.2%

**Table 6 ijerph-17-07265-t006:** Abnormal Score vs. Actual Abnormal ratio.

Abnormal Score	Actual Abnormal/Samples
Top 0.01%	24/500
Top 20.01%–Top 20.02%	2/500
Top 40.01%–Top 40.02%	3/500
Top 60.01%–Top 60.02%	2/500
Top 80.01%–Top 80.02%	1/500
Random 0.01% except top 1%	1/500, 2/500, 2/500

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
