# Peer review of "Medical Fraud and Abuse Detection System Based on Machine Learning"

_ijerph, 2020, doi:10.3390/ijerph17197265_

Round 1

Reviewer 1 Report

The authors describe the domain area (medical fraud and abuse in health care systems) well and used enough related resources of the problem domain. Manuscript clearly stated the problem they are going to deal with which is detecting fraudulent medical treatments or services to the patient. Motivation of research is clearly stated and also authors used enough analysis of the data being used for training the model. However, the manuscript should be further improved, below are few suggestions.    

  • The related work section should be improved and add one paragraph that clearly stated the distinguishing features of this study over existing studies. Also, there are very few related work and references discussed in the manuscript, include some recent articles such as:

    - Ijaz, Muhammad Fazal, Muhammad Attique, and Youngdoo Son. "Data-Driven Cervical Cancer Prediction Model with Outlier Detection and Over-Sampling Methods." Sensors10 (2020): 2809.
    - Ali, Farman, Shaker El-Sappagh, SM Riazul Islam, Amjad Ali, Muhammad Attique, Muhammad Imran, and Kyung-Sup Kwak. "An intelligent healthcare monitoring framework using wearable sensors and social networking data." Future Generation Computer Systems (2020).
  • Check the numbering of Sections, I find there is problem in that, check Section 2.1.
  • The machine learning techniques should be explained further so it will help to understand author’s level of understanding ( The authors claim that they used neural network model  for quantifying the relation but didn’t describe the designed neural network model in detail 
  • The formulas and notations in the paper are poorly explained.
  • The figures and diagrams (page 5, 6 …) have no proper numbering. Its very confusing to use only ‘figure’ in caption section and they all lack enough explanation
  • It would be better to use more machine learning model for comparing results in addition to neural networks. State vector machine as an example and more work on explaining the accuracy
  • Regarding the data, try to find and come up with more labeled and a bigger data as it is playing a key role in building a perfect model.

Author Response

Point 1: [The related work section should be improved and add one paragraph that clearly stated the distinguishing features of this study over existing studies. Also, there are very few related work and references discussed in the manuscript]

Response 1: We newly added more related work and references, and analyzed where we are better, on page 3, line 95-114.

Point 2: [Check the numbering of Sections, I find there is problem in that, check Section 2.1.] and [The figures and diagrams (page 5, 6 …) have no proper numbering. Its very confusing to use only ‘figure’ in caption section and they all lack enough explanation]

Response 2: We newly normalized and formatted all figure numbers and table numbers, and specified the titles of figures:

Figure 1: The top10 disease record count distributions.   showed on page 4, line 141

Figure 2: The disease record count distributions.  showed on page4, line 147

Figure 3: System Diagram. showed on page 6, line 200

Figure 4: (a) the t-SNE result by K-means, (b) the result by isolation forest. showed on page 7, line 213

Figure 5: ranks of known abnormal record based on (a) our model (b) traditional rule showed on page 7, line 227

Table 1: main data structure in the digital healthcare system showed on page 3, line 119

Table 2: mainly features showed on page 5, line 153

Table 3: Multilabel algorithms' result showed on page 6, line 194

Table 4: Detection rate of different algorithms showed on page 7, line 224

Table 5: Abnormal Score vs. Actual Abnormal ratio showed on page 8, line 241

Point 3: [The machine learning techniques should be explained further so it will help to understand author’s level of understanding ( The authors claim that they used neural network model  for quantifying the relation but didn’t describe the designed neural network model in detail]

Response 3: We newly further explained our neural network model, on page 4, line 149; page 5, line 169-172. In addition, we will add a structure diagram of the model later.

Point 4: [The formulas and notations in the paper are poorly explained.]

Response 4: We newly explained the formulas and notations in the paper, on page 5, line 173-178, 181-183.

Point 5: [It would be better to use more machine learning model for comparing results in addition to neural networks. State vector machine as an example and more work on explaining the accuracy]

Response 5: We newly compared several outlier detection algorithms, on page 6, line 184-186, l194

Point 6: [Regarding the data, try to find and come up with more labeled and a bigger data as it is playing a key role in building a perfect model.]

Response 6: We take your suggestion on obtaining/constructing new data very seriously, but because of the privacy and security issues involved, data acquisition requires the approval of the government and relevant agencies, which is much more complicated. Therefore, we have to use methods such as a focal-loss function or up and down sampling to mine information from existing data and maintain label balance as much as possible.

downsample and upsample: on page 4, line 142-147

focal-loss function: fomula (2), introduced on page 5, line 173-178

Besides, because our study-work lives are too busy, we lack enough time and energy to obtain more data and enhance the model to adapt to imbalanced data classification (although it is enough for now). But we also consider them essential, so we put them in the future work section ( page 9, line 296-302 )

Reviewer 2 Report

The paper looks at an AI method of analysing health records for anomalies around expenditure and drug use to isolate fraud or abuse. This is a good objective internationally

There are some minor expression errors in Introduction, and some more explanation of method might help:

May be worth explaining the "health care settlement scheme"

The process is well described and the difficulty, but also the link to a final human analysis is a buffer against most concerns raised

Very thorough description of the analysis and modelling used

I understand from section 2.0 you are using unsupervised learning with the neural network

However it is not clear where your clusters were developed from. The disease types? or the characteristics developed from advice by the analysts

Also not clear how the anomalies you identified related to these characteristics you developed. 

The conclusion is good, and I am wondering if the behaviour relates more to patient and doctor than disease, so cover more of the cases than you fear?

Author Response

Point 1: [explaining the 'health care settlement scheme'] and [link to a final human analysis is a buffer against most concerns raised]

Response 1: As the real environment is often more complicated than the simulated data or partial data, and we cannot access the whole data we need to well tune the model because of the privacy and security issues involved, our model, unfortunately, can only assistant, not replace, the human analysis, at this moment. Hence we are just offering an auxiliary tool, not a gross settlement scheme.

Point 2: [not clear where your clusters were developed from. The disease types? or the characteristics developed from advice by the analysts?]

Response 2: Yes, the features we designed were mainly based on analysts' experiences, including personal information, history healthcare behavior, etc. This section was newly described on page 4, line 149, with a features table.

Point 3: [not clear how the anomalies you identified related to these characteristics you developed]

Response 3: We generated features based on the analysts' advice and then selected the top 20 by the mutual information score on the labelled dataset, which was  newly described on page 4.

Point 4: [I am wondering if the behaviour relates more to patient and doctor than disease, so cover more of the cases than you fear?]

Response 4: As mentioned, we can hardly tackle these problems directly, facing the privacy and security issues involved. Instead, we implemented a focal-loss function and upsampling/downsampling techniques to solve the data noise problem and the data category imbalance problem. And definitely, we discussed more efforts that deserve enduring in the Future Work section, including the application for more data and trying other balance skills.

Round 2

Reviewer 1 Report

Authors have adequately revised the manuscript and addressed reviewer comments. I don't have any further comments.

Author Response

Thanks for your help.